# Climate variables are not the dominant predictor of Arctic shorebird distributions

**Christine M. Anderson**[1]*, **Lenore Fahrig**[1], **Jennie Rausch**[2], **Paul A. Smith**[3]

**1** Department of Biology, Geomatics and Landscape Ecology Laboratory, Carleton University, Ottawa, ON, Canada, **2** Canadian Wildlife Service, Environment and Climate Change Canada, Yellowknife, NT, Canada, **3** Wildlife Research Division, Environment and Climate Change Canada, Ottawa, ON, Canada

* christineanderson3@carleton.ca

**Data Availability Statement:** All data and code files are available from the Zenodo database (https://doi.org/10.5281/zenodo.7613304).

**Funding:** PAS - P57 - ArcticNet - https://arcticnet.ulaval.ca PAS - 2017-06391 - National Science and

## Abstract

Competing theoretical perspectives about whether or not climate is the dominant factor influencing species' distributions at large spatial scales have important consequences when habitat suitability models are used to address conservation problems. In this study, we tested how much variables in addition to climate help to explain habitat suitability for Arctic-breeding shorebirds. To do this we model species occupancy using path analyses, which allow us to estimate the indirect effects of climate on other predictor variables, such as land cover. We also use deviance partitioning to quantify the total relative importance of climate versus additional predictors in explaining species occupancy. We found that individual land cover variables are often stronger predictors than the direct and indirect effects of climate combined. In models with both climate and additional variables, on average the additional variables accounted for 57% of the explained deviance, independent of shared effects with the climate variables. Our results support the idea that climate-only models may offer incomplete descriptions of current and future habitat suitability and can lead to incorrect conclusions about the size and location of suitable habitat. These conclusions could have important management implications for designating protected areas and assessing threats like climate change and human development.

## Introduction

Habitat suitability models are used to describe the relationship between species' presence in geographic space and the characteristics of the environment, and to estimate how likely a species is to occupy unsampled locations. Along with closely related species distribution models, environmental niche models, resource selection functions and bioclimatic envelope models [but see 1], these analyses are increasingly popular for addressing a wide range of questions, such as how species will respond to climate change [e.g. 2], identifying priority areas for threatened species [e.g. 3], managing landscapes [e.g. 4], and understanding the spread of invasive species [e.g. 5]. However, different theoretical perspectives about how these relationships between species and their environment are structured have important consequences for the variables included in these models, and therefore how we use them to address conservation problems.

Engineering Research Council of Canada - https://www.nserc-crsng.gc.ca/index_eng.asp PAS and JR - Institutional funding - Environment and Climate Change Canada - https://www.canada.ca/en/environment-climate-change.html CA - 518225-2018 - National Science and Engineering Research Council of Canada - https://www.nserc-crsng.gc.ca/index_eng.asp CA - W. Garfield Weston Award in Northern Research (Doctoral) 2019 – 2020 - The W. Garfield Weston Foundation. - https://westonfoundation.ca/northern-science-and-knowledge/ The funders had no role in study design, data collection and analysis, decision to publish, or preparation of the manuscript.

**Competing interests:** The authors have declared that no competing interests exist.

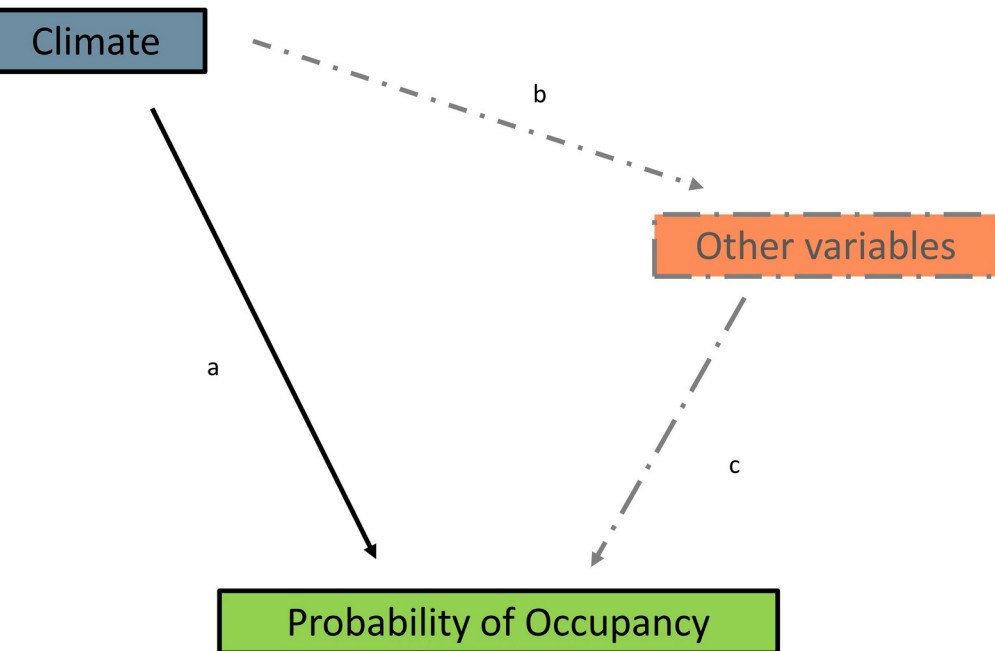

**Fig 1. Conceptual relationships between the drivers of species occupancy.** Illustration of the common assumption that, over large spatial extents, climate predictors drive the probability of species occupancy. Such climate-only models imply that the total effects of climate predictors, i.e., both their direct effects (a) and their indirect effects through their effects on additional predictors such as land cover (b*c), are much greater than the direct effects of these additional predictors (c). The underlying assumption is that climate is the main driver of variation in the predictors that influence species occupancy, and so it is only necessary to include climate predictors in species occupancy models.

One hypothesis suggests that climate is the dominant factor influencing species' distributions at large spatial scales [6, 7]. Climate influences species' distributions directly, for example through thermal tolerances, and indirectly, for example through climate's influence on land cover (Fig 1). Interestingly, many of the papers that invoke this hypothesis are studies of future species distribution based on climate-only models [i.e. 8]. One reason for this practice is pragmatic: often there are no future predictions available for the other potential covariates of distribution such as land cover [9]. However, these papers also often argue that at large spatial extents, such as entire ranges for widespread species, the direct and indirect effects of climate encompass the majority of the influences on species' distributions [10]. 'Climate-only' models are often evaluated using a threshold, such as a model with an AUC of at least 0.7 has "good" explanatory power, and usually climate-only models exceed the threshold [i.e. 11, 12].

A competing hypothesis suggests that habitat suitability models need to be more comprehensive because there can be additional important factors shaping species distributions that may not be well described as an indirect effect of climate, as is often assumed in climate-only models. In contrast to the future-oriented models described above, papers estimating current species distributions often include a wide range of predictors, many of which are non-climatic, relating instead to resources and habitat structure [13]. Studies aimed at estimating current habitat suitability typically compare multiple models to identify a best model, which is rarely a climate-only model [i.e. 14–16]. In particular, biotic interactions may be important even at large spatial extents, not just at local scales [17]. For example, many species require the presence of specific plants as food or hosts to facilitate their presence in a region [18]. Precipitation is often a poor proxy for the water available to plants, as topography and soil substrate control how precipitation translates into soil moisture [19]. Soil predictors such as pH and nutrients

are important predictors of plant distribution, are strongly influenced by underlying geology, irrespective of climate [20].

In this paper, we ask how much do variables in addition to climate help to explain habitat suitability? To do this we model species occupancy using path analyses, which allow us to estimate the indirect effects of climate on other predictor variables, such as land cover. We also use partitioning to quantify the total relative importance of climate versus additional predictors in explaining species occupancy. We addressed our question using data on the distribution of Arctic-breeding shorebirds. Most papers studying habitat suitability for Arctic-breeding shorebirds have been conducted at smaller extents, and for the most part they do not include climate variables, likely because climate does not vary enough over smaller extents to be a relevant predictor [i.e., 21–23]. The few large-extent models that have been used to predict habitat suitability for these species include a model that largely relies on climate predictors [24], but also a report suggesting that additional predictors beyond climate were important for predicting current and future occupancy of Red Knot (*Calidris canutus)* and Semipalmated Sandpipers (Calidris pusilla) [25]. Based on previous studies of Arctic shorebird habitat associations, the additional predictors we consider are land cover, snow cover, substrate chemistry, and elevation, and the standard deviation of elevation, as shorebirds prefer flat habitats [21, 24–27].

*A priori*, we expected that the probability of shorebird occupancy at sites distributed over large spatial extents can be estimated considerably better by explicitly including additional variables in habitat suitability models, rather than assuming that these additional variables are themselves driven by climate. If climate is the dominant factor influencing the occupancy of breeding shorebirds across the Arctic, we expect that the variation in shorebird occupancy is mostly explained by the combination of the direct effects of predictors from climate-only models and their indirect effects through their effects on the additional predictors. If, independent of climate, additional predictors have important influences on the occupancy of breeding shorebirds across the Arctic, we expect that a substantial amount of the explained variation in shorebird occupancy is uniquely attributed to the additional predictors not found in climate-only models.

## Methods

### Overview

To look at the predictors of shorebird occupancy, their interrelationships, and their relative effect sizes, we developed a hypothetical path diagram outlining our hypotheses about how climate predictors and additional environmental predictors are interrelated (Fig 2). We identified predictors that have been important in other models of habitat suitability for Arctic-breeding shorebirds [21, 24–27]. We used path analysis to verify the structure of our path diagram and assess the degree to which the effects of additional predictors are driven by climate. We then used a deviance partitioning approach to test whether the total (direct + indirect) effects of climate predictors explained most of the variation in shorebird occupancy. All analyses were performed using R Statistical Software v4.2.1 [28].

### Shorebird surveys

Plots were surveyed for shorebirds across the Canadian Arctic as part of the Arctic Program for Regional and International Shorebird Monitoring (PRISM) [29], which covers all of Arctic North America as defined by the Circumpolar Arctic Vegetation Map [30]. The Canadian PRISM data used here includes 2336 plots across the Canadian Arctic (Fig 3), an area of 3.5 million km$^2$. Given this massive area, surveys were completed over a number of years, from

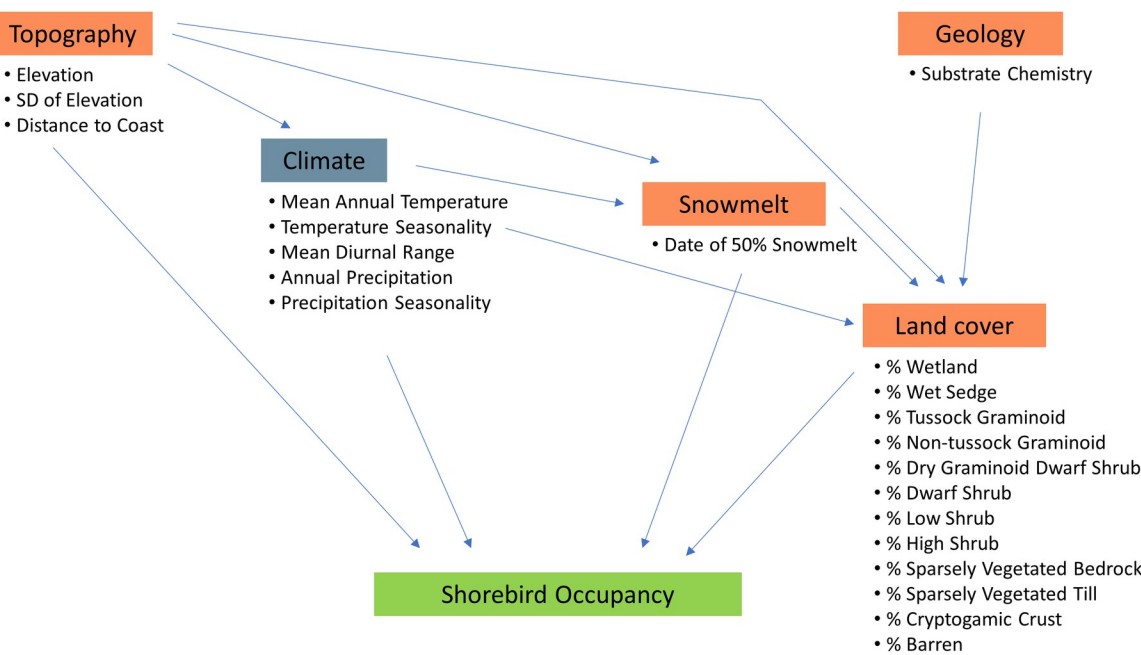

**Fig 2. Path diagram of the hypothesized relationships among predictors of shorebird occupancy.** Groups of predictors with a blue label are included in climate-only models, and groups of predictors with an orange label are additional predictors included in climate +additional models. Arrows indicate direct causal paths between all combinations of variables in the respective groups are included in the path model of occupancy for each shorebird species. For example, the arrow from climate to land cover represents 60 arrows, one from each climate variable to each land cover variable.

1994–2018, with a majority of the sampling effort concentrated between 2003 and 2018. The Canadian Arctic was divided into 12 regions based on logistical considerations. Within each region plots were selected by random sampling, stratified by habitat. The proportion of plots in wet, moist and dry habitats varied by region depending on habitat abundance [see 29 for more details], but on average 16%, 29%, and 56% of plots were in wet, moist, and dry habitats respectively. In all years, surveys were conducted between June 18 and July 15. This corresponds with the latter part of shorebird courtship and the early part of incubation, when breeding territories could be identified from the birds' territorial displays. Plots were typically 12–16 ha (400 m x 300–400 m). We excluded one set of surveys because extensive flooding led to very atypical breeding conditions. We also excluded surveys for which observers recorded nearby human development, which is rare across the study area and could potentially alter shorebird habitat preferences. For 211 plots that were visited in more than one year, we randomly selected one of the surveys to be included in the final dataset.

Plots were surveyed for breeding birds following PRISM protocols [29]. Two surveyors searched the plot walking straight-line transects, covering a breadth of 50 m with the observers situated 25 m apart, using a GPS to ensure complete coverage of the entire plot. Surveys took approximately 90 mins. Observers recorded the number and species of all birds observed within each plot. Altogether we observed 19 species of shorebirds, 17 of which were observed in at least 30 plots, our minimum threshold for developing habitat suitability models: American Golden-Plover (*Pluvialus dominica*), Baird's Sandpiper (*Calidris bairdii*), Black-bellied Plover (*Pluvialis squatrola*), Buff-breasted Sandpiper (*Tryngites subruficollis*), Dunlin (*Calidris alpina*), Least Sandpiper (*Calidris minutilla*), Pectoral Sandpiper (*Calidris melanotos*), Red Knot (*Calidris cantus*), Red Phalarope (*Phalaropus fulicarius*), Red-necked Phalarope (*Phalaropus lobatus*), Ruddy Turnstone (*Arenaria interpres*), Sanderling (*Calidris alba*), Semipalmated

Sandpiper (*Calidris pusilla)*, Stilt Sandpiper (*Calidris himantopus*), Wilson's Snipe (*Gallinago delicata*), and White-rumped Sandpiper (*Calidris fuscicollis*). In our analyses we chose to model occupancy instead of density because shorebirds were observed at very low densities (average of <5 birds/survey for all 17 species, ranging from 0.03–0.68 birds for each individual species), with a frequent counts of 0 (no birds were observed in 54% of surveys). Surveys were observational, and involve no capture or sampling. Lands surveyed include Federal Crown Lands, Inuit Owned Lands, and Federal and Territorial Protected Areas. Access to these lands was granted through permits to JR including: Environment and Climate Change Canada (NWT-MBS-17-03, MM-NR-2022-NU-005, NUN-NWA-17-04, SC-NR-2001-NT-004, SC-NR-2022-NU-005), Government of Nunavut (WL2022-031), Kitikmeot Inuit Association (KTX115X007), Kivalliq Inuit Association (KVX15N05), Qikiqtani Inuit Association (Q14X001).

## Environmental predictors

We used two categories of environmental predictors of shorebird occupancy: predictors found in 'climate-only models', and additional predictors in 'climate+additional models'. For the climate predictors we used the bioclimatic data available from WorldClim 2 at a 30 arc-second resolution (~1km) [31]. This data is typical of that used in climate-only habitat suitability models because it is global in scope and easily accessible. We eliminated any predictors that had correlations higher than 0.7, selecting the 5 uncorrelated climate predictors that seemed most relevant to shorebird occupancy: annual mean temperature, temperature seasonality (standard deviation of mean monthly temperature x 100), mean diurnal range (mean of monthly max temp–min temp), annual precipitation, and precipitation seasonality (coefficient of variation of monthly precipitation).

Additional predictors in the climate+additional models of shorebird occupancy included topography, geology, snowmelt, and land cover variables. The topography predictors we included were elevation, the standard deviation of elevation, and distance to coast. We used elevation data available from WorldClim 2, also at a 30 arc-second resolution (~1km). From this, we derived the standard deviation of elevation over a 5km grid. We calculated distance to coast from coastline data available from Natural Earth [32]. To represent geology, we used substrate chemistry data from the Circumpolar Arctic Vegetation map [30], which includes 3 categories: acidic (pH < 5.5), circumneutral (pH 5.5–7.2) and carbonate (pH > 7.2). Snowmelt timing was derived from an 8-day composite MODIS product, and we used the day of 50% snowmelt per 500m pixel [33]. We used land cover data from the Circa-2000 Northern Land Cover Map of Canada. This dataset derives 15 land cover classes from Landsat data 1999–2002 at a 30m resolution [34].

## Statistical analysis

Before building models of habitat suitability for each species of shorebird, we first determined the spatial extent at which day of snowmelt and each land cover predictor most strongly affected each species, their "scale of effect" [35]. Habitat selection is scale dependent, therefore identifying the appropriate scale at which species respond is important for making correct inferences about the relative importance of environmental variables [36]. We did not complete this step for climate, topography or substrate because these variables were measured at much larger spatial resolutions, and show little variation at small spatial resolutions. To find the scales of effect, we first filtered the plots to a reduced set that were at least 10km from each other. We calculated the mean day of snowmelt and the proportion of each land cover type at multiple scales, centered on the middle of each plot. We tested spatial scales from a radius of

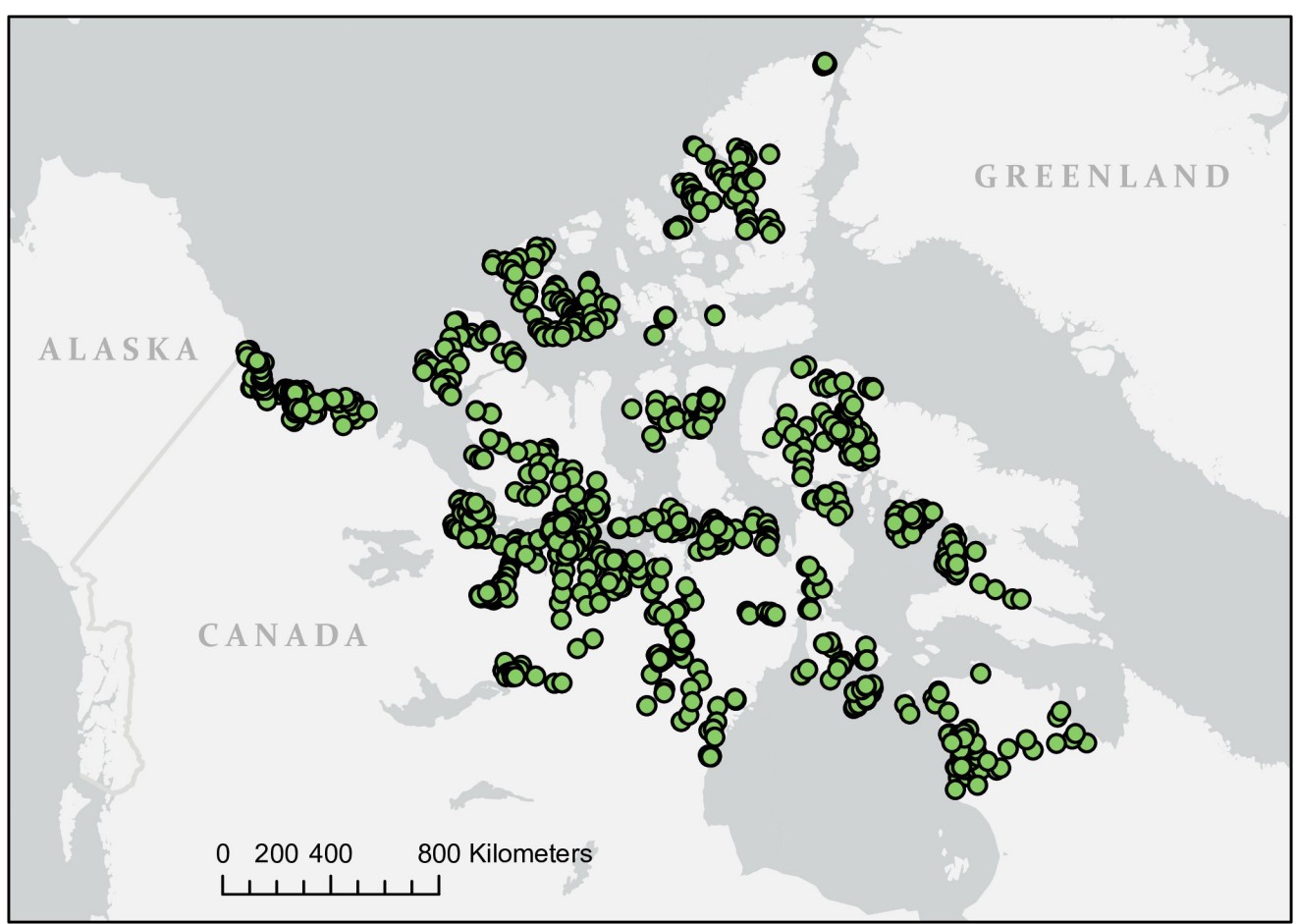

**Fig 3. Map of survey locations.** 2336 plots were surveyed for breeding shorebirds across northern Canada, 1994–2018. Reprinted from ArcGIS under a CC BY license, with permission from Esri, original Copyright 2022 Esri (Basemaps supported by Esri, HERE, Garmin, OpenStreetMap contributors, and the GIS User Community).

200m, approximately covering the plot itself, to 10km, increasing by increments of 100m up to 2500m and by increments of 500m thereafter. We created a series of logistic regression models relating the occupancy of each of the 17 shorebird species to each predictor at each spatial scale. Then for each predictor, we identified the scale of effect for that species as the extent where the model had the lowest AIC value. In the models described below, we entered the snowmelt timing and land cover predictors at their scales of effect.

We used path analysis to test whether the effects of the additional variables on habitat suitability for shorebirds were largely explainable as indirect effects of climate. Path models were fit using the R package piecewiseSEM [37]. We fit a path model for each of the 17 shorebird species, following the causal structure illustrated in Fig 2. We also included correlated error terms among all climate variables, and among all land cover variables. All models included an offset for plot size. We used a test of directed separation (d-sep) to evaluate the goodness-of-fit for our hypothesized path model, assessing the conditional independence between pairs of variables in the model. Significant d-sep test results indicated associations between substrate and climate, and substrate and shorebird presence, so we included these links as correlated error terms as well. For each species model, the median standardized effects and 95% confidence intervals were estimated using nonparametric bootstrapping from 10,000 resamples using the

R package semEff [38]. The path coefficients we present are semi-partial correlations, which are the correlation between the unique components of predictors (residualised on other predictors) and the response variable. Note that this approach controls for multicollinearity between predictors, whereas many path analyses present standardized partial regression coefficients which do not control for multicollinearity, which often creates an upward bias in perceived effect sizes [39]. Model fit was assessed with Fisher's *C* statistic, for which a significant value indicates that no potentially significant missing paths were excluded, as well as visually using plots of residuals calculated from the median coefficient values. To assess the relative strength of individual predictors, we compared the absolute values of their total effects on shorebird species occupancy. The total effect of a predictor is the sum of its direct effects (the path coefficients between the predictor and the shorebird response) and its indirect effects (the product of the path coefficients between the predictor, intermediate predictors, and the shorebird response). Effects were considered to be significant when 95% confidence intervals did not contain 0.

To assess the overall relative contribution of climate-only model predictors and additional predictors to shorebird occupancy, we used a deviance partitioning approach. We used Generalized Linear Models (GLMs) with a binomial distribution to determine the deviance explained by climate and additional predictors. Species were analysed separately. Deviance was partitioned, using the ratio of the null deviance and the deviance explained by one of the following factors: climate predictors, additional predictors, deviance shared between climate and additional predictors, and unexplained deviance.

## Results

We observed 11,636 shorebirds in surveys of 2336 plots from 1994–2018. At least one shorebird was observed on 54% of plots. By species, occupancy varied from 1% (Red Knot) to 14% (Semipalmated Sandpiper); the average (±SD) species occupancy was 6 ± 4%. Descriptive statistics of the environmental predictor values are in S2 Table. The mean scale of effect for land cover classes was 4400m (median = 3000m). For virtually all land cover classes, the scale of effect varied from the smallest possible scale (200m, just covering the plot) to the largest possible scale (10km) depending on the species. Similarly, each species responded to the different land cover classes at very different spatial scales (S3 Table). The median scale of effect for snowmelt timing was 5000m (median = 3500m).

Direct effects of climate-only model predictors accounted for on average 17% of the total explained deviance in shorebird occupancy (Fig 4). The total effect of climate-model predictors (direct effect + shared/indirect effect) accounted for on average 43% of the total explained deviance in shorebird occupancy. The unique effects of additional predictors included in climate+additional models, beyond the indirect effects of climate through these predictors, accounted for on average 57% of the total explained deviance in shorebird occupancy (Fig 4). Depending on the species, the ratio of deviance explained by climate-only predictors vs additional predictors ranged from 80% climate/20% additional to 10% climate/90% additional. The mean explained deviance for our models of shorebird occupancy was 27% (ranging from 9%-42%). The mean AUC of our models of shorebird occupancy was 0.87 (ranging from 0.72–0.93). The full results of the GLMs used for deviance partitioning can be found in S4 Table.

Overall, the total effect sizes of climate predictors (direct + indirect sizes; Fig 2) and additional predictors on shorebird occupancy were similar (Fig 5). Our path model included the paths illustrated in Fig 2 (Fisher's *C* statistic = 3.14, P = 0.21). The 5 predictors with the largest significant absolute effect on shorebird occupancy were additional predictors, specifically the predictors that described the proportion land cover at the species-specific scale of effect of

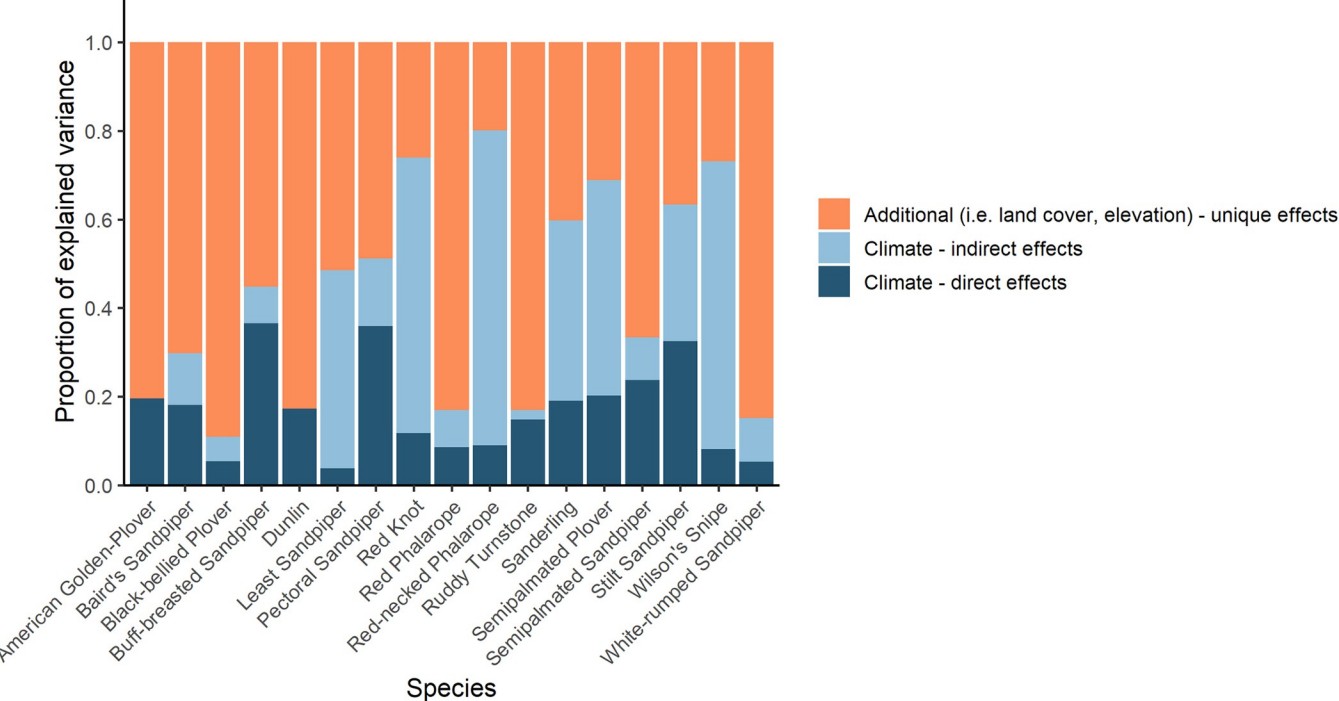

**Fig 4. Deviance partitioning for GLMs predicting the occupancy of each of 17 shorebird species.** Each bar represents the total deviance explained for each species. The direct effects of climate, the proportion of explained deviance attributed to predictors from the climate-only model, is dark blue. The indirect effects of climate, the proportion of deviance that is explained by both climate and additional variables, is light blue. The total effect of climate on the occupancy of each species is the sum of the dark and light blue segments. The proportion of explained deviance attributed only to the additional predictors from the climate+additional models is orange. For half of the species shown, the unique effects of the additional variables explain more deviance in shorebird occupancy than the total effect of climate.

high shrub, dwarf shrub, sparsely vegetated till, dry graminoid dwarf shrub, and non-tussock graminoid. The climate predictors with the largest significant absolute effect on shorebird occupancy were mean diurnal range and mean annual temperature (Fig 5). A summary of the significant total effects of predictors on shorebird occupancy can be found in S5 Table. The full results of the 17 species-specific path models, including summaries of the direct, indirect and total effects of the paths between all variables can be found in S6 Table.

## Discussion

Our results suggest that the direct and indirect effects of climate variables are not enough to explain the range-wide variation in the occupancy of Arctic-breeding shorebirds. Individual land cover variables are often stronger predictors than the direct and indirect effects of climate combined. In models with climate+additional variables, on average the additional variables accounted for more than half of the explained deviance, independent of shared effects with the climate variables. Our results support the idea that climate-only models may offer incomplete descriptions of current and future habitat suitability, and can lead to incorrect conclusions with important management implications.

Even at broad spatial scales, Arctic-breeding shorebirds typically have distinct land cover preferences that are stronger predictors of habitat suitability than climate, and we show here that land cover is not a simple derivative of climate. Aside from climate, tundra vegetation and soils are also the product of the geological parent material and water drainage [40]. Acidic and non-acidic tundra soils have substantially different biogeochemical fluxes, which in many

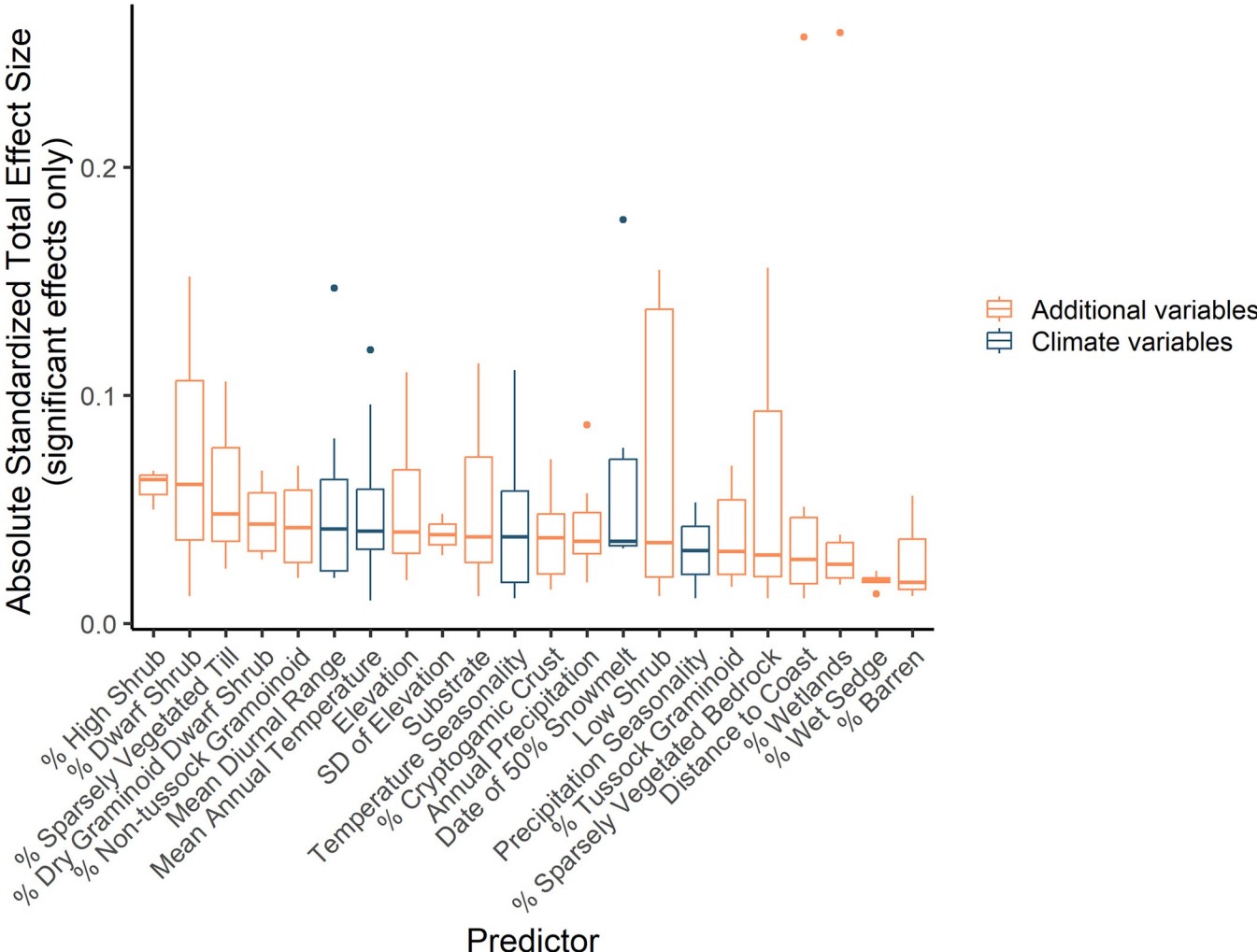

**Fig 5. Total effect sizes of significant paths.** Box and whisker plots showing the median and interquartile range of the absolute total effect sizes of significant (p>0.05) paths from individual path models of occupancy for the 17 most abundant shorebird species in the Canadian Arctic. Survey data was collected from 1994–2018 (Fig 3). The total effect sizes for each variable are the sum of its direct and indirect effects on shorebird occupancy (Fig 2). The predictors with the five largest effect sizes were additional variables. The magnitude of the effect sizes for climate and additional variables fall in a similar range.

cases are larger differences than between bioclimatic vegetation zones [40]. These land cover and vegetation characteristics, which are only partly predicted by climate, are well established predictors of shorebird habitat. For example, Dunlin and Red Phalaropes are more likely to be present in in moist, lowland habitats [21], while Red Knots prefer sparsely vegetated habitats [26]. Habitat suitability models for eight shorebird species on the Alaska North Slope showed that all species were much less likely to be breed in upland shrub habitats [27]. Semipalmated Plovers commonly nest on stony shorelines, which helps them avoid nest predation by Arctic Foxes [41].

We should expect a spectrum of how well habitat suitability for individual species can be predicted by climate. For example, Pöyry et al [42] found that butterflies with high mobility were modelled less accurately by climate-only models than species with low mobility. Species that use a wide range of climates are generally modelled less accurately than species found in a limited range of climatic conditions [43]. The shorebirds studied here breed in the Arctic, and winter in tropical and subtropical areas, encountering vastly different climatic conditions

 

throughout their annual cycle. Here, we saw that within closely related shorebird taxa in the same Arctic ecosystem, the total effect of climate explained anywhere from 10% to 80% of the explained deviance in shorebird occupancy.

In the debate about whether climate-only models are adequate for describing habitat suitability, one important factor that leads different papers to come to different conclusions is the spatial scale at which studies are conducted [44]. Even Pearson and Dawson [10], advocates of climate-only models, recommend that these analyses be undertaken with careful consideration: ". . .bioclimate envelope models can provide a useful starting point when applied to suitable species and **at appropriate scales**." However, it seems that whether or not a climate-only model is appropriate for the given species and scale is often assumed rather than tested. For example, Wauchope et al. [24] write "Data were interpolated to a spatial grain size of 10 x 10 km to reflect the approximate resolution of most of the distributional records and a scale at which climate, rather than microhabitat factors, is more likely to be limiting [45]".

Identifying which predictors are relevant at a particular spatial scale requires consideration of both the grain and the extent [44]. Spatial grain has been demonstrated to influence the relative importance of climate predictors. For example, Luoto et al. [47] found that climate-only models of bird distributions in Finland were improved by including land-cover at 10km and 20km resolutions, but not at 40km and 80km resolutions. The spatial extent of a study also plays a significant role in defining how much variability we find in each of the predictor variables. The extent of our study covers only a part of the breeding range of some species, and just beyond the range of others. Studies with much larger extents, covering the full range of climate conditions across the continent would have a stronger signal for climate, identifying a wider range of conditions species do not use. Habitat selection is a hierarchical process [46]. Identifying the appropriate spatial scale for the type of habitat selection being studied will help identify when climate-only models are sufficient, and when other predictors such as land cover are important.

Optimizing the scale of effect for land cover and other predictors that can be measured at a range of spatial scales is an important step that is often neglected [36]. Here, we found that Arctic-breeding shorebirds were responding to land cover at a radius of up to 10km, but that the scale of effect differed across species and land cover categories. A single-scale study of vertebrates in Florida [47], contrary to our results, found that including additional predictors had a relatively minor effect on the accuracy of climate-only habitat suitability models. They suggested that the 4km grain of their study may have been too coarse to resolve the land cover associations for their study species, following the hypothesis that biotic interactions are most important at smaller spatial scales. Another important difference between our study and that of Bucklin et al. [48] may be that the spatial extent of our study area was approximately 20x larger; therefore, there may be more available variation in land cover across our sites, increasing its importance. Single-scale studies may therefore miss important predictors, because they are not measuring the predictors at the scale at which species are responding to the predictor [44].

Other reasons that climate-only models may be preferred are purely practical. Although important, additional predictors are often neglected because they are unavailable, or harder to obtain at the extent and resolution needed [20, 49]. Particularly for future predictions of habitat suitability, future climate projections are widely available, while other future projections such as land cover are much more limited in their availability, their spatial resolution, their accuracy, and the level of detail that they provide in terms of the number of classification categories [50, 51].

We conclude that environmental predictors beyond climate are important for improving the accuracy of habitat suitability models explaining species distributions. We found that a

 

large portion of explained deviance in shorebird occupancy is related to direct effects of additional predictors, such as elevation and land cover, and that these effects are not well captured as indirect effects of climate. It is therefore unwise to assume that climate models are always sufficient for explaining habitat suitability. By making these assumptions, we place unnecessary limits on our understanding of species relationships with their environment. Incorrect conclusions from habitat suitability models could have important management implications for designating protected areas and assessing threats like climate change and human development. As many countries seek to expand their protected areas networks to meet international commitments under the Convention on Biological Diversity [52], a more complete understanding of species' distributions can have important management implications now and especially in the future, under scenarios of climatic change.

## Supporting information

**S1 Fig. Maps of northern Canada showing where each species of breeding shorebirds was present during PRISM surveys 1994–2018.** Reprinted from ArcGIS under a CC BY license, with permission from Esri, original Copyright 2022 Esri (Basemaps supported by Esri, HERE, Garmin, OpenStreetMap contributors, and the GIS User Community).
(PDF)

**S2 Fig. AIC values for all tested radii to determine the optimal scale of effect at which land cover and snow cover affect shorebird occupancy.**
(PDF)

**S1 Table. Years in which each PRISM region was surveyed.** See [29] for map of survey regions.
(DOCX)

**S2 Table. Range of values for environmental predictors.**
(DOCX)

**S3 Table. Radius (m) of the optimal scale of effect at which land cover and snow cover affect shorebird occupancy, determined by AIC.**
(DOCX)

**S4 Table. Full results of GLMs used for deviance partitioning.**
(TXT)

**S5 Table. Path analysis: Summary of the significant total effects of predictors on shorebird occupancy.**
(CSV)

**S6 Table. Path analysis: Full results of the 17 species-specific path models, including summaries of the direct, indirect and total effects of the paths between all variables.**
(CSV)

## Acknowledgments

We thank the field crews who conducted shorebird surveys for the PRISM program, and the Polar Continental Shelf program for providing logistical support.

## Author Contributions

**Conceptualization:** Christine M. Anderson, Lenore Fahrig, Paul A. Smith.

**Data curation:** Christine M. Anderson, Jennie Rausch.

**Formal analysis:** Christine M. Anderson.

**Funding acquisition:** Jennie Rausch, Paul A. Smith.

**Investigation:** Christine M. Anderson.

**Methodology:** Christine M. Anderson, Paul A. Smith.

**Project administration:** Christine M. Anderson, Jennie Rausch, Paul A. Smith.

**Resources:** Jennie Rausch, Paul A. Smith.

**Supervision:** Lenore Fahrig, Paul A. Smith.

**Validation:** Christine M. Anderson.

**Visualization:** Christine M. Anderson.

**Writing – original draft:** Christine M. Anderson.

**Writing – review & editing:** Christine M. Anderson, Lenore Fahrig, Jennie Rausch, Paul A. Smith.

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
