## [Decision Letter · Decision Letter 0]

28 Mar 2023

PONE-D-23-03560Climate variables are not the dominant predictor of Arctic shorebird distributionsPLOS ONE

Dear Dr. Anderson,

Thank you for submitting your manuscript to PLOS ONE. After careful consideration, we feel that it has merit but does not fully meet PLOS ONE’s publication criteria as it currently stands. Therefore, we invite you to submit a revised version of the manuscript that addresses the points raised during the review process.

We look forward to receiving your revised manuscript.

Kind regards,

Tunira Bhadauria, Ph.D.

Academic Editor

PLOS ONE

Journal Requirements:

"The funders had no role in study design, data collection and analysis, decision to publish, or preparation of the manuscript.

PAS - P57 - ArcticNet - https://arcticnet.ulaval.ca

PAS - 2017-06391 - National Science and Engineering Research Council of Canada -  https://www.nserc-crsng.gc.ca/index_eng.asp

PAS and JR - Institutional funding - Environment and Climate Change Canada - https://www.canada.ca/en/environment-climate-change.html

CA - 518225-2018 - National Science and Engineering Research Council of Canada -  https://www.nserc-crsng.gc.ca/index_eng.asp

CA - W. Garfield Weston Award in Northern Research (Doctoral) 2019 – 2020 - The W. Garfield Weston Foundation. - " ext-link-type="uri" xlink:type="simple">https://westonfoundation.ca/northern-science-and-knowledge/"

3. We note that Figure 3 and S1 Figure in your submission contain [map/satellite] images which may be copyrighted. All PLOS content is published under the Creative Commons Attribution License (CC BY 4.0), which means that the manuscript, images, and Supporting Information files will be freely available online, and any third party is permitted to access, download, copy, distribute, and use these materials in any way, even commercially, with proper attribution. For these reasons, we cannot publish previously copyrighted maps or satellite images created using proprietary data, such as Google software (Google Maps, Street View, and Earth). For more information, see our copyright guidelines: http://journals.plos.org/plosone/s/licenses-and-copyright.

a. You may seek permission from the original copyright holder of Figure 3 and S1 Figure to publish the content specifically under the CC BY 4.0 license.  

Reviewers' comments:

Reviewer's Responses to Questions

**Comments to the Author**

1. Is the manuscript technically sound, and do the data support the conclusions?

Reviewer #1: Yes

Reviewer #2: Yes

Reviewer #3: Yes

2. Has the statistical analysis been performed appropriately and rigorously? 

Reviewer #1: Yes

Reviewer #2: Yes

Reviewer #3: Yes

3. Have the authors made all data underlying the findings in their manuscript fully available?

Reviewer #1: Yes

Reviewer #2: Yes

Reviewer #3: Yes

4. Is the manuscript presented in an intelligible fashion and written in standard English?

Reviewer #1: Yes

Reviewer #2: Yes

Reviewer #3: Yes

5. Review Comments to the Author

Reviewer #1: Manuscript entitled" Climate variables are not the dominant predictor of Arctic shorebird distributions" are suitable for publication with some minor revision

1."climate-only models may offer incomplete descriptions of current and future habitat suitability, and can lead to incorrect conclusions with important management implications" are important finding.

2.Abstract may be expanded by adding application of work

3.future prospective may be strengthen

4.A abstract figure must be draw to represent methodology and results

Reviewer #2: In general I really like what you have done here. I do think there are a few areas where you could clarify your methods. In addition, I think you missed a key point about why your study found greater support for habitat variables versus climate compared to other studies. Specifically, your sampling does not extend far south of the breeding ranges of the species in your analyses (unlike a lot of SDM papers which mine continental or global data sets). This limits both the climate variation and the number of structural zeros in your data. In other analyses/studies, these extra zeros inflate the importance of the climate signal. Your discussion is short, so you can add text to address that point. I include a marked up copy of the manuscript where I make specific recommendations.

Reviewer #3: I have read the manuscript. This is a good paper. The idea and statistical analysis and text written well.

However, I would like to know the answer of authors regarding the following noise of using of landcover variable: You used the distribution records of 1994-2018, right? but the landcover variable of which year has been used? it is not clear in the manuscript. Here my main question is that how did you cover the noise of period for landcover variable? If you say for a clear year like 2018, I ask again how was your landcover for previous years as it is impacted by human pressures and climate variables in this period!

My another question is that about the correlation between landcover and climate variables ! if the climate variables influence on the landcover , so how it can be justified by correlation analysis?

Please correct "Error! Reference source not found"

Please insert clear objective in the text.

6. PLOS authors have the option to publish the peer review history of their article (what does this mean?). If published, this will include your full peer review and any attached files.

Reviewer #1: **Yes: **Kuldip jayaswall

Reviewer #2: **Yes: **Steven L. Van Wilgenburg

Reviewer #3: **Yes: **Hossein Mostafavi

quillbot-extension-portal/quillbot-extension-portal

---

## [Author Response · Author response to Decision Letter 0]

13 Apr 2023

Journal Requirements

We have updated the manuscript to meet PLOS ONE’s style requirements, by updating file naming, title capitalization, and figure labeling. 

"The funders had no role in study design, data collection and analysis, decision to publish, or preparation of the manuscript.

PAS - P57 - ArcticNet - https://arcticnet.ulaval.ca

PAS - 2017-06391 - National Science and Engineering Research Council of Canada - https://www.nserc-crsng.gc.ca/index_eng.asp

PAS and JR - Institutional funding - Environment and Climate Change Canada - https://www.canada.ca/en/environment-climate-change.html

CA - 518225-2018 - National Science and Engineering Research Council of Canada - https://www.nserc-crsng.gc.ca/index_eng.asp

CA - W. Garfield Weston Award in Northern Research (Doctoral) 2019 – 2020 - The W. Garfield Weston Foundation. - https://westonfoundation.ca/northern-science-and-knowledge/"

As you can see at the beginning of the financial disclosure statement copied above, we had originally included this statement as required. We have moved it to the end of this statement in our resubmission. 

3. We note that Figure 3 and S1 Figure in your submission contain [map/satellite] images which may be copyrighted. 

We require you to either (1) present written permission from the copyright holder to publish these figures specifically under the CC BY 4.0 license, or (2) remove the figures from your submission.

The base maps used in Figure 3 and S1 Figure come from our licensed copy of ArcGIS Pro. We investigated the copyright requirements for publishing these base maps. Here is a copy of the master agreement that cover this copyright: https://www.esri.com/content/dam/esrisites/en-us/media/legal/ma-full/ma-full.pdf and here is a summary of some of the relevant information from the master agreement: https://doc.arcgis.com/en/arcgis-online/reference/static-maps.htm. This summary states that ArcGIS base maps are permitted to be used in academic publications, with attribution included near the map. Thus, we have added an attribution statement to the captions of Figure 3 and S1 Figure, following the following ArcGIS citation guide (https://support.esri.com/en-us/knowledge-base/faq-what-is-the-correct-way-to-cite-an-arcgis-online-ba-000012040) and an example of another paper which has been published in PLOS ONE (https://journals.plos.org/plosone/article/figures?id=10.1371/journal.pone.0220941). 

Reviewer #1

Manuscript entitled" Climate variables are not the dominant predictor of Arctic shorebird distributions" are suitable for publication with some minor revision

1."climate-only models may offer incomplete descriptions of current and future habitat suitability, and can lead to incorrect conclusions with important management implications" are important finding.

2.Abstract may be expanded by adding application of work

We have included information in our abstract and discussion adding more detail about what the management implications would be of inaccurate habitat suitability models (lines 36-37 and 364-365)

3.future prospective may be strengthen

We have added more detail to our discussion in response to Reviewer #2’s comments. This new paragraph, lines 339-351 addresses how future studies should identifying the appropriate spatial scale for the type of habitat selection being studied in order to clarify why certain type of predictors are relatively more or less important.

4.A abstract figure must be draw to represent methodology and results

My first impression is that the reviewer is asking for a graphical abstract, which is not listed as one of the requirements for an abstract in PLOS ONE. If the reviewer is asking a more general figure to represent the methodology and results, I would suggest that Figure 1 illustrates the conceptual basis for our paper, Figures 2 and 3 illustrate the methodology, and Figures 4 and 5 illustrate the results. 

Reviewer #2

In general I really like what you have done here. I do think there are a few areas where you could clarify your methods. In addition, I think you missed a key point about why your study found greater support for habitat variables versus climate compared to other studies. Specifically, your sampling does not extend far south of the breeding ranges of the species in your analyses (unlike a lot of SDM papers which mine continental or global data sets). This limits both the climate variation and the number of structural zeros in your data. In other analyses/studies, these extra zeros inflate the importance of the climate signal. Your discussion is short, so you can add text to address that point. I include a marked up copy of the manuscript where I make specific recommendations.

Line 145: Excluding this is likely fine for this analysis, but could you could also have included to first see if it was an outlier. Might have been informative?

Perhaps. From our perspective, we chose to exclude this year because our observation was that such an extreme amount of the habitat that would normally be available was under water that it did not seem reasonable to be associating the presence or absence of birds with the landcover data we used. 

Line 158: If highest breeding evidence is relevant in this analysis, then add details of the codes/categories or provide a citation. However, if not, then delete this text.

I suppose it isn’t very relevant, because we accepted any evidence of breeding. We have removed this line from the text. 

Line 192: You might want to address the variation in resolution b/w your different variables. 30 arc-seconds is circa 1km

We have added this clarification

Line 216: What about time of season?

The surveys were intentionally done at a time of year when there would be a small enough variation in the occupancy and behaviour of the birds that we feel comfortable not including this as an offset. 

Line 222: I assume the semi-partial coefficients are within a bootstrap, but are you basing inference on the median (and 95%CI) across bootstraps?

That’s right. We’ve added clarification to this effect on line 225. 

Line 227: Were the residuals calculated from observed - prediction from median coefficient values? Or did you build a distribution across all 10K bootstraps? Why not additional measure of predictive accuracy on independent withheld data?

The residuals were calculated from the median coefficient values. We didn’t include an independent measure of predictive accuracy on independently withheld data for the path analysis because methods to do this with path analysis and structural equation models aren’t well established yet. This is a fairly cutting edge are of research in statistical journals (i.e. de Rooij, M., Karch, J. D., Fokkema, M., Bakk, Z., Pratiwi, B. C., Kelderman, H. (2022). SEM-based out-of-sample predictions. Structural Equation Modeling: A Multidisciplinary Journal, 1-17). We were able to calculate the bootstrapped semi-partial coefficients and effects using the r packages piecewiseSEM and semEFF, but unfortunately these packages aren’t yet designed to work with packages like predict that would allow us to calculate independent measures of predictive accuracy. 

We did feel as though the statistical description of our path model was lacking. We have now included a Fisher’s C statistic, which is a measure of goodness of fit, indicating that there were no potentially significant paths missing.

Line 244: Very low densities (5 birds / survey across 17 spp). In methods you might wish to mention that this is why you are modeling this occupancy and not as counts 

Yes, precisely. Good idea. We have added a description of this in the methods lines 169-72.

Line 247: AIC selection of scale effect would be good to have in the as supplemental material tables

Good idea. We have added this: S2_Fig

Line 250: I don't see an S3 Fig. Only S1 Fig.

We have corrected this typo. We were referring here to S3 Table, not Fig. 

Line 295: I am generally inclined to agree with your approach and conclusions. However, another difference to consider relative to other studies, yours still has fairly restricted sampling and thereby excludes a lot of climate variation that entails structural zeros. I think a lot of studies have strong climate signals because sampling goes FAR beyond the species extant ranges (e.g. for example if you included eBird and BBS with your PRISM data, or if you extent PRISM several hundred KM into the boreal forest) that add climate noise to the data, but are truly structural zeros because they are not even plausible as possible localities of occurrence. Sampling between the 49th parallel and your southernmost location would add a lot of zeros and climate variation. Since a lot of studies mine global datasets, they contain those excess zeros and the climate signal is huge. I am not sure you can fully resolve this here, but should speak to it somewhat as an alternative reason other studies find climate dominates. Perhaps simulation studies perhaps will be needed to resolve how best to approach the issue (should a threshold prob of occurrence be reached before trying to partition climate versus habitat?). Perhaps other papers should consider limiting the distances from known breeding range to better attribute deviance between climate and habitat? Maybe including those wider data to model climate envelopes of species ranges, but restricting spatial extent of contributing data to partition climate vs. habitat is the answer?

This is definitely something we were thinking about when we were doing our analyses, and I’m glad you pointed this out. This point was missing from our discussion. We’ve added a new paragraph from lines 339–351 to address these ideas. 

Line 314: is 'to' out of place here?

Yes. We have corrected this typo. 

Reviewer #3: 

I have read the manuscript. This is a good paper. The idea and statistical analysis and text written well.

However, I would like to know the answer of authors regarding the following noise of using of landcover variable: You used the distribution records of 1994-2018, right? but the landcover variable of which year has been used? it is not clear in the manuscript. Here my main question is that how did you cover the noise of period for landcover variable? If you say for a clear year like 2018, I ask again how was your landcover for previous years as it is impacted by human pressures and climate variables in this period!

In our description of the landcover variable, we mentioned that the data is circa the year 2000. We have added some extra details to help clarify – this dataset was created with Landsat data from 1999-2002. In a perfect world, it would be nice to have landcover data for each year, but the Arctic is a particularly data poor region, so having 30m resolution data from close to the middle of our survey period is as good as we can get. In the Arctic, change in land cover caused by humans occurs at a tiny fraction of what occurs in other regions, therefore we do not expect that land cover would have varied significantly across the region during this time period. 

My another question is that about the correlation between landcover and climate variables ! if the climate variables influence on the landcover , so how it can be justified by correlation analysis?

I’m not quite sure what this question is asking. Yes, climate has an influence on landcover, therefore they are correlated. This is why we used path analysis and variance partitioning, specifically to address this issue. 

Please correct "Error! Reference source not found"

This has been corrected. 

Please insert clear objective in the text.

Our objective was stated in line 84: “ how much do variables in addition to climate help to explain habitat suitability?”. We have now prefaced this sentence saying “In this paper…” to make it more clear that this is the main objective.

---

## [Editor Report · Decision Letter 1]

16 Apr 2023

Climate variables are not the dominant predictor of Arctic shorebird distributions

PONE-D-23-03560R1

Dear Dr. Anderson 

We’re pleased to inform you that your manuscript has been judged scientifically suitable for publication and will be formally accepted for publication once it meets all outstanding technical requirements.

Kind regards,

Tunira Bhadauria, Ph.D.

Academic Editor

PLOS ONE

Additional Editor Comments (optional):

Reviewers' comments:

quillbot-extension-portal/quillbot-extension-portal

---

## [Editor Report · Acceptance letter]

25 Apr 2023

PONE-D-23-03560R1 

Climate variables are not the dominant predictor of Arctic shorebird distributions 

Dear Dr. Anderson:

I'm pleased to inform you that your manuscript has been deemed suitable for publication in PLOS ONE. Congratulations! Your manuscript is now with our production department. 

Kind regards, 

on behalf of

Dr. Tunira Bhadauria 

Academic Editor

PLOS ONE